# Analysis of the Actions of Net Zone Approach in Padel: Validation of the NAPOA Instrument

**DOI:** 10.3390/ijerph19042384

**Published:** 2022-02-18

**Authors:** Adrián Escudero-Tena, Diego Muñoz, Javier García-Rubio, Sergio J. Ibáñez

**Affiliations:** 1Training Optimization and Sport Performance Research Group (GOERD), Sport Science Faculty, University of Extremadura, 10005 Caceres, Spain; jagaru@unex.es (J.G.-R.); sibanez@unex.es (S.J.I.); 2Department of Musical, Plastic and Corporal Expression, Faculty of Sport Sciences, University of Extremadura, 10003 Caceres, Spain; diegomun@unex.es

**Keywords:** racket sports, game analysis, content validity, reliability, notational analysis, NAPOA

## Abstract

To carry out research that analyzes performance indicators through observational methodology, it is necessary to have validated tools. The purpose of this study was to design and validate a tool to ascertain the characteristics of the strokes that padel pairs use to reach the net and their consequences in the two subsequent shots of the game. Eleven experts, who had to meet four of the five inclusion criteria established by the researchers, participated in the process. Aiken’s V coefficient and confidence intervals were used to analyze content validity and Cronbach’s α coefficient to calculate reliability. The adequacy and wording of the sixteen variables initially designed were evaluated. Four variables were eliminated due to obtaining values <0.90 in Aiken’s V coefficient in adequacy. The rest of the variables were modified in their wording, according to the qualitative evaluations of the experts, or were considered correct. The reliability of the instrument was acceptable (α = 0.89). The NAPOA instrument is novel, as it is of interest to analyze the characteristics of the strokes that padel players use to achieve the offensive position.

## 1. Introduction

Padel has gone from being a minority sport to being one of the most practiced sports in the world, as it is played in more than 40 countries. This growth has led to an increase in men’s, women’s, team, senior, and junior championships. Likewise, the number of sports facilities, federated clubs, sponsors, or licenses around the world has increased [1]. Specifically, the most established professional men’s and women’s circuit in the world is the World Padel Tour (WPT), a competition that is based in Spain, but which organizes tournaments in different countries each season.

Interest in this sport has also been seen in the increase in scientific publications. There are many areas in which studies have been carried out on padel in recent years—educational [2], anthropometric [3,4], physiological [5,6], psychological [7,8], etc.

Specifically, there is special interest in the analysis of performance indicators in padel. Studies have been conducted to describe the competition [9], the technical–tactical actions that may be more effective [10,11,12], physical condition [13,14], movements [15,16], biomechanics [17,18], or the discovery of game indicators [19,20,21]. In addition, various investigations in padel have studied the differences that exist between winning and losing pairs [21,22,23,24] or between women’s and men’s padel [12,25,26].

Research related to the analysis of performance in padel has concluded that there are two basic playing areas. The net area, which is the one in which the pair plays in positions close to the net, and the background area, which is the one in which the pair plays at the back of the court [27]. In both of these zones, offensive and defensive shots could be played. However, pairs that win games in padel perform more attack actions (near the net) in 85% of the points, spend more time in the net area, and hit fewer shots from the back of the court during the game [21,22,23]. In addition, these studies show that about 80% of the winners are obtained from near the net. Thus, there is a relationship between scoring points and occupying areas close to the net. While the objective of the pair of players who are in the background zone is to fight to achieve the net position, the objective of the players who are in the net zone is to fight to preserve it [23].

Various studies have shown that the lob is the most used technical–tactical action by men and women padel pairs from the background position to achieve the net position. However, the point does not end, thus allowing continuity in the game and giving rise to more exchanges of position between pairs during the same point [11,28,29]. To obtain these results, the researchers used ad hoc tools, based on observational methodology, which allow the notational recording of the game actions studied. However, no designed and validated observational tool has been found aimed at studying the shots used by padel partners to achieve the offensive position, despite the fact that there are several observational tools that have been designed and validated aimed at the study of padel based on the analysis of the quantitative and qualitative judgment of a group of experts [30,31,32].

After a review of the scientific literature, the non-existence of a validated instrument that analyzes the shots used by padel pairs to achieve the net position or reach the net was confirmed. Therefore, the objective of this research was to design and validate an observation instrument to ascertain the characteristics of the different strokes that padel pairs use to reach the net and their consequences in subsequent shots.

## 2. Materials and Methods

### 2.1. Research Design

The design was classified within the instrumental methodology, ex post facto [33], to develop and validate an observation tool to assess the characteristics of the different strokes that padel pairs use to get to the net and their consequences in subsequent shots.

### 2.2. Participants

The participants were chosen deliberately and intentionally, since expert subjects were sought who were capable of transmitting knowledge and information about the object of study, as well as making evaluations which could provoke reflection and help researchers [34]. In addition, an attempt was made to select a group of experts who met the inclusion criteria established by the researchers: (i) to possess a Ph.D, (ii) to possess the federative qualification as a trainer in padel and/or in another racket sport, (iii) to teach or have taught at university, (iv) to have publications with a theme oriented to the analysis of the game of padel, and (v) to work or have worked as a padel coach or coach of another racket sport. Eighteen experts who were considered by the researchers to meet the inclusion criteria were invited to participate. Finally, the sample that participated in the validation of the instrument consisted of eleven experts, who had to meet four of the five established inclusion criteria. Thus, nine experts met the five inclusion criteria and only two met four (Table 1).

### 2.3. Study Variables

In addition to the variables that made up the instrument, variables were identified to analyze their content validity and reliability. Content validity is defined as the degree to which a variable adequately represents the instrument [35]. In this study, the technique used to achieve an optimal level of content validity was the assessment based on the criteria of the experts [36]. The experts assessed the adequacy and wording of each variable through a quantitative scale from 1 to 10. Adequacy is the extent to which a variable is considered relevant to form part of the tool and wording refers to a variable being correctly written. Likewise, the experts made a qualitative assessment if they deemed it appropriate. Moreover, reliability, understood as the internal reproducibility of a measure [35], was analyzed using Cronbach’s α coefficient.

### 2.4. Instrument

An instrument was designed consisting of contextual variables, which define the game, the players or the state of play, and specific variables that analyze the game actions that are intended to be assessed. These variables were defined based on their categorical core and their degree of openness [37].

Among the variables that describe the game situation is the difference between the pair that wins the match and the pair that loses it, in order to know if the shot that the padel pairs use to reach the net and its consequence is related to achieving success. Various investigations highlight the importance of occupying and maintaining a position close to the net to increase the chances of success [22,38].

In padel, it is important to know the position of the player on the court, depending on whether they are on the left or right side. Previous research indicates that the way the players who position themselves on the left side act on the court is different from that of the players on the right side [39,40,41].

The data obtained in other investigations suggest that the performance profile of padel players differs according to their hand dominance [39,41,42], therefore this tool also takes laterality into account.

The pair that serves during a game has a significant advantage over the returning pair, especially in the first seconds of the game [43,44], for that one instrument variable is punch status. Whether the pair is serving or returning may influence the shot that is used to reach the net.

The partial result of the game, the set, or the match are also items to be included in this tool. It is very interesting to know what the players do according to whether they are winning, losing, or tying, since various investigations have shown that players act differently according to the score [11,28,45].

Another item that is included in this tool is the key point, as there are several studies that have analyzed the key points in padel [28,31]. They suggest that players use longer rest times before points that can affect the score (key points), which could be directly related to physiological (recovery), tactical, and psychological factors, due to the importance of these points.

Although no studies have been found that analyze the streak in padel, it has been studied in other sports [46]. The number of points previously won or lost can affect the way the next point is played, and therefore this variable was included in the instrument.

The lob is the technical–tactical action most used by padel players to reach offensive positions. However, the point does not end, giving rise to more exchanges of position between pairs during the same point [11,28,29]. In this instrument, several variables have been developed, taking into account the aforementioned investigations: specifically, the variables hitting area, type of shot, direction of the shot, action of the rival pair, and action at the net in order to record the characteristics of the shot that the padel pairs use to reach the net and its consequence.

Finally, the variable order of the rally was introduced, to discover the moment during the point in which the shot occurs that the padel pairs use to reach the net and the number of shots per point, a variable that has been studied in padel by various researchers [47,48]. These studies show an average of about 9–10 shots per point.

Taking these variables into account, a first version of the instrument was developed. The initial definition of the 16 designed variables, their categorical core and the opening range for the observational analysis tool, the characteristics of the different strokes used by padel pairs to reach the net, and their consequence in subsequent strokes (Net Approach in padel observational analysis, NAPOA) are presented in Table 2.

### 2.5. Process

Once the analysis of the literature was carried out, the problem statement was identified and a tool was built that would analyze the characteristics of the different shots that padel pairs use to achieve the net and its consequence in subsequent strokes. Once the variables and categories were defined, the researchers selected a group of experts who met the inclusion criteria they had established. Upon the response of the experts, the data were recorded in an Excel sheet. Quantitative data were used to calculate content validity through Aiken’s V coefficient and confidence intervals and reliability from Cronbach’s α coefficient. Qualitative data were used to improve the final wording of the instrument.

### 2.6. Analysis of Data

Content validity was calculated using Aiken’s V coefficient [49], which is used to quantify the relevance of a variable with respect to a group of experts. The value of Aiken’s V coefficient ranges between 0.00 and 1.00, thus, the closer the value is to 1.00, the more agreement there will be among the experts regarding the content evaluated. For its calculation, the Visual Basic 6.0 software developed by Merino and Livia [50] was used, which uses the formula modified by Penfield and Giacobbi [51], where $\bar{X}$ refers to the mean of the scores obtained by the experts, ı is the lowest value on the scale (1), and K is its range (10 − 1 = 9).
(1)$$V=\frac{\bar{X}-ı}{K}$$


This application allows obtaining the confidence intervals at the 95% and 99% levels using the score method [51]. This confidence interval calculation is a confirmatory test that shows greater goodness for the creation of instruments designed for the first time [50].

The initial formula proposed by Aiken [49] was followed to establish the criteria for elimination, modification, or acceptance of variables, applying the central limit theorem. Although the number of variables (m) and number of experts (n) was less than twenty-five, the range of the scale (c) was greater than seven. In this calculation proposal, z = significant value of content validity; m = number of variables; n = number of experts, and c = range of the scale.
(2)$$V=\frac{z}{0.2\sqrt{\frac{3\mathrm{mn}\left(c-1\right)}{\left(c+1\right)}}}+0.5$$


The criteria used by other researchers were followed when validating instruments, establishing the cut-off point to eliminate an item at 95% confidence. When the values were between 95% and 99% of confidence, the items should be improved. An item is considered to be correctly designed when it has a confidence value greater than 99% [52,53]. It is a highly demanding criterion for the validation of a tool. Therefore, in the present investigation, variables with mean values lower than 0.90 in Aiken’s V (below 95% confidence) were eliminated, variables with mean values between 0.90 and <1.00 (between the 95% and 99% confidence) were modified, and the variables with mean values at 1.00 (greater than 99% confidence) were considered correct (Table 3).

Cronbach’s α coefficient [54] was used to analyze the reliability of the instrument. This coefficient is used to check if the instrument being evaluated collects faulty information and therefore would lead to wrong conclusions or, on the other hand, if it is a reliable instrument that makes stable and consistent measurements. Thus, [55] shows that an acceptable reliability is considered from 0.70, although other authors indicate that it would be more advisable to obtain values above 0.80 [56,57]. Statistical analysis was performed with SPSS v.21 software (IBM Corp. 2012. IBM SPSS Statistics for Windows, NY: IBM Corp., Armonk, USA).

## 3. Results

Table 4 shows the results obtained using Aiken’s V coefficient and their confidence intervals regarding adequacy.

It was observed that variables 8 (key point), 9 (streak), 15 (order of the rally), and 16 (rally) did not exceed the critical value for Aiken’s V with respect to the adequacy that was established at 0.90, and therefore these variables were eliminated from the record sheet.

Table 5 shows the results obtained after calculating Aiken’s V coefficient and its confidence intervals regarding the wording.

The experts stated that the variables 2 (player), 5 (partial result of the game), 8 (key point), 9 (streak), 11 (type of shot), 13 (action of the rival pair), 14 (action in the net zone), 15 (order of the rally), and 16 (rally), should be revised. None of them exceeded the critical value for Aiken’s V with respect to the wording, and therefore, special treatment was necessary with these variables to improve them.

Table 6 shows, by way of example, the qualitative assessments provided by the experts and the actions that were carried out accordingly.

Once all the changes and consequential corrections of the quantitative (Aiken’s V) and qualitative (contributions) assessment of the experts had been made, the tool was validated and is presented in Appendix A.

Finally, Table 7 shows the values for the reliability of the tool through Cronbach’s α coefficient, before and after eliminating the variables that obtained a value lower than 0.90 in Aiken’s V coefficient with respect to adequacy. After eliminating the 4 variables (key point, streak, order of the rally, and rally) suggested by the experts, Cronbach’s α coefficient improved.

## 4. Discussion

To carry out research that analyzes performance indicators through observational methodology, it is necessary to have validated tools. The objective of this research was to design and validate an observation tool to analyze the characteristics of the different strokes that padel pairs use to reach the net and their consequences in subsequent shots. Thus, an instrument was created—the NAPOA, made up of 12 variables, which allows us to analyze these game situations that constantly occur in padel. Despite the fact that these game situations have been the object of study of various investigations [11,28,29] in different game contexts (amateur padel, professional women’s padel, or professional men’s padel), an instrument that analyzes them in a specific way, built from the analysis of the quantitative and qualitative judgment of a group of experts, has not been established so far. This is surprising, since there are several observational tools that have been designed and validated aimed at the analysis of the game in padel [30,31,32].

For the validation of an instrument to be satisfactory, a series of guidelines must be met [36,58], which will be developed throughout this discussion: (i) the selection criteria of experts; (ii) the number of experts that comprise the panel; (iii) the procedure used by the experts to assess the validity of content; (iv) the statistical or quantitative procedures to evaluate the experts’ scores; and (v) the selection criteria used to determine whether the items are kept, modified, or eliminated from the final proposal to be included in the instrument.

The selection criteria for the experts were custom-defined for the present investigation. Except for one who is in the process, all the experts are Ph.D.s, thus guaranteeing their scientific training. Likewise, all the experts have taught at university and are authors of scientific publications where the object of study is the analysis of the game in padel. Moreover, except for one, all the experts have the federative qualification and have worked as a padel coach or that of another racket sport, guaranteeing their experience. Other investigations, aimed at the validation of tools, have used selection criteria similar to those described. That is, they have used Ph.D.s [31,53,59], experts with scientific publications related to the topic to be analyzed [52,53], and experts who have federal qualifications and have served as coaches [30,31]. In addition, the rule that experts must meet 80% or more of the inclusion criteria to be part of the sample has been used by other investigations on this topic [31,52]. Therefore, the quality of the experts participating in the study is guaranteed, as are their quantitative and qualitative assessments.

In the sports field, ten or more subjects offer an acceptable estimate for the content validity of a validation instrument [59,60,61]. The sample of this study is made up of eleven experts, so this requirement has been exceeded. Thus, the contributions of our experts are sufficient in terms of numbers for the validation of this observation tool.

The experts made a quantitative assessment of each of the variables in the NAPOA instrument. This assessment awarded scores from 1 to 10 for the drafting and adequacy of the items, as carried out in other studies [31,52], and the procedure that was used to quantitatively analyze the content validity of the tool has been used in other investigations [31,62,63,64]. This procedure uses the cut-off point to eliminate a variable at 95% confidence. Thus, four of the sixteen variables were eliminated (key point, streak, order of rally, and rally), since they obtained values lower than 0.90 in Aiken’s V for adequacy. The experts considered that these variables should not be part of the tool. On the other hand, when the values were between 95% and 99% of confidence, the variables were improved. Specifically, the variables player, partial result of the game, type of shot, action of the rival pair, and action at the net were modified. The comments made by the experts were taken into account for their modification, since they carried out a qualitative assessment of the variables, which was essential for the final development of the tool [36,65]. The degree of openness per player on the right side and player on the left side was changed in the player variable. In the partial result of the game, the categories of the opening range advantage-40 and 40-advantage should be omitted when the instrument is used for the analysis of World Padel Tour matches. The ranges of the variables type of shot (lob, passing, and chiquita), action of the rival pair (winner, error, and continuity) and action in the net zone (winner, error, continuity, and no action) were modified. Finally, an item was considered to be correct when it had a value greater than 99% confidence [52,53], in other words, when Aiken’s V was 1.00. Thus, a new proposal of the validated tool was built, which was made up of 12 items, both situational—defining the state of play, and specific—analyzing the stokes that padel pairs use to reach the offensive position and their consequences in both subsequent shots (Appendix A).

Various studies aimed at the validation of observational tools use the same procedure that was used in this research to obtain reliability [31,63,66,67]. The tools in these investigations, like the instrument in this study, reach optimal reliability values, since they all obtain values higher than those that the experts mark as a reference [56,57]. In addition, it is novel to calculate said reliability once the variables that reached values lower than 0.90 in Aiken’s V coefficient in adequacy had been eliminated, since its value improved considerably, from 0.84 to 0.89 in Cronbach’s α coefficient. Thus, NAPOA has sufficient internal consistency, that is, the variables measure the constructs of the characteristics of the strokes used by padel pairs to achieve the net and their consequences consistently.

## 5. Conclusions

The tool designed in this study is valid. Although a very high cut-off point was determined due to the number of variables and experts, all the variables that make up the final tool (Appendix A) present an appropriate value in Aiken’s V coefficient with respect to adequacy. In addition, the wording of the variables that presented a value between 0.90 > 1.00 in Aiken’s V coefficient was modified according to the qualitative evaluations of the experts.

NAPOA is a reliable tool, since the value obtained in Cronbach’s α coefficient is very high and the variables of the instrument consistently measure the characteristics of the strokes that the padel pairs use to reach the net and their consequences in the two strokes.

This instrument is valuable and very useful for other researchers who face the possibility of carrying out this type of study. In addition, it is important to use validated and reliable observation tools to analyze the analysis of the game in padel.

This tool makes it possible to ascertain the characteristics of the strokes used by the padel pairs to reach the net and their consequence in the two subsequent shots. It would be convenient to use this tool for future scientific studies in all kinds of contexts, that is, in different padel sports categories and in both men’s and women’s matches. It would be of great help for padel players to know which shot is the most suitable or effective to achieve the offensive position, its characteristics, and what the consequences would be. Likewise, this information is vital for padel coaches, and for the development of training tasks and game strategies.

## Figures and Tables

**Table 1 ijerph-19-02384-t001:** Inclusion criteria met by experts.

	Experts										
	1	2	3	4	5	6	7	8	9	10	11
Criterion 1	x	x	x	x	x	x	x	x	x	x	
Criterion 2	x	x	x	x	x	x		x	x	x	x
Criterion 3	x	x	x	x	x	x	x	x	x	x	x
Criterion 4	x	x	x	x	x	x	x	x	x	x	x
Criterion 5	x	x	x	x	x	x	x	x	x	x	x

x = meets inclusion criteria.

**Table 2 ijerph-19-02384-t002:** Category system of the NAPOA tool.

Variables	Description	Degree of Opening
1. Pair	Pair of the player who makes the stroke to reach the net depending on the final result of the match	1. Pair that wins the match
2. Pair that loses the match
2. Player	Position of the player on the court who makes the stroke used by the padel pair to reach the net	1. Drive
2. Reverse
3. Laterality	Dominant hand of the player who makes the stroke that the padel pair uses to reach the net	1. Right-handed
2. Left-handed
4. Service status	Defines if the partner of the player who makes the stroke to reach the net is serving or returning	1. Returning pair
2. Serving pair
5. Partial game result	Partial result of the game of the pair of the player who makes the stroke to reach the net	1. 0–0	8. 0–30	15. 30–40
2. 15–0	9. 30–30	16. 40–40
3. 0–15	10. 40–0	17. 40-advantage
4. 15–15	11. 0–40	18 Advantage-40
5. 30–15	12. 40–15	19. Tie-break
6. 15–30	13. 15–40	
7. 30–0	14. 40–30	
6. Partial set result	Partial result of the set of the pair of the player who makes the shot to reach the net	1. 0–0	12. 1–5	23. 5–3
2. 1–0	13. 2–2	24. 3–5
3. 0–1	14. 3–2	25. 4–4
4. 1–1	15. 2–3	26. 5–4
5. 2–1	16. 4–2	27. 4–5
6. 1–2	17. 2–4	28. 5–5
7. 3–1	18. 5–2	29. 6–5
8. 1–3	19. 2–5	30. 5–6
9. 4–1	20. 3–3	31. 6–6
10. 1–4	21. 4–3	
11. 5–1	22. 3–4	
7. Partial match result	Partial result of the match of the pair of the player making the stroke to reach the net	1. 0–0
2. 1–0
3. 0–1
4. 1–1
8. Key point	Points that could have an impact on the result of the match, in which either pair had the option of winning a game, set or match	1. Yes
2. No
9. Streak	Defines whether the pair of the player who made the stroke to reach the net won or lost the previously played point (s)	1. Won the previous point
2. Won the 2 previous points
3. Won the 3 previous points or more
4. Lost the previous point
5. Lost the 2 previous points
6. Lost the 3 previous points or more
7. First point of the match
10. Hitting zone	Area from which the stroke is made that the padel pair uses to reach the net	1. 1a
2. 2a
3. 3a
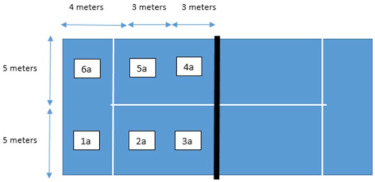	4. 4a
5. 5a
6. 6a


11. Shot type	Stroke used by the padel pair to reach the net	1. Lob
2. No lob
12. Direction of the shot	Path taken by the ball once it has been hit by the player who makes the stroke used by the padel pair to reach the net	1. Parallel
2. Cross-court
13. Rival pair action	Define the consequence of the shot made by the rival pair	1. Winner
2. Forced error
3. Unforced error
4. Continuity
14. Action in the net zone	Define the consequence of the first shot that the pair makes at the net	1. Winner
2. Forced error
3. Unforced error
4. Continuity
5. No action
15. Rally order	Moment during the point at which the shot is made that the padel pair uses to reach the net	1. Very soon (2nd–6th shot)
2. Soon (7th–11th shot)
3. Normal (12th–16th shot)
4. Late (17th–21st shot)
5. Too late (22nd or more shots)
16. Rally	Number of shots during the point	1. Very short (2–8 shots)
2. Short (9–16 shots)
3. Normal (17–24 shots)
4. Long (25–32 shots)
5. Very long (33 or more shots)

**Table 3 ijerph-19-02384-t003:** Criteria to follow for the acceptance, modification, or elimination of the variables.

		Wording		
		1.00	[0.90–<1.00]	<0.90
Adequacy	1.00	Correct	Wording is modified	Wording is modified
	[0.90–<1.00]	Adequacy is modified	Adequacy and wording are modified	Adequacy and wording are modified
	<0.90	It is eliminated	It is eliminated	It is eliminated

**Table 4 ijerph-19-02384-t004:** Results of Aiken’s V coefficient and confidence intervals (Adequacy).

Variables	Adequacy						
	Mean	Aiken’s V	95% Confidence Interval			99% Confidence Interval	
			Lower Limit		Upper Limit	Lower Limit	Upper Limit
1	10	1.00		0.96	1.00	0.93	1.00
2	9.91	0.99		0.94	0.99	0.92	0.99
3	10	1.00		0.96	1.00	0.93	1.00
4	10	1.00		0.96	1.00	0.93	1.00
5	10	1.00		0.96	1.00	0.93	1.00
6	10	1.00		0.96	1.00	0.93	1.00
7	10	1.00		0.96	1.00	0.93	1.00
8	8.64	0.85	*	0.76	0.90	0.73	0.91
9	8.36	0.82	*	0.73	0.88	0.70	0.89
10	10	1.00		0.96	1.00	0.93	1.00
11	10	1.00		0.96	1.00	0.93	1.00
12	10	1.00		0.96	1.00	0.93	1.00
13	10	1.00		0.96	1.00	0.93	1.00
14	9.91	0.99		0.94	0.99	0.92	0.99
15	8.00	0.78	*	0.68	0.84	0.65	0.86
16	8.55	0.84	*	0.75	0.89	0.72	0.91

* <0.90.

**Table 5 ijerph-19-02384-t005:** Results of Aiken’s V coefficient and confidence intervals (Wording).

Variables	Wording						
	Mean	Aiken’s V	95% Confidence Interval			99% Confidence Interval	
			Lower Limit		Upper Limit	Lower Limit	Upper Limit
1	10.00	1.00		0.96	1.00	0.93	1.00
2	9.00	0.89	*	0.81	0.93	0.78	0.94
3	10.00	1.00		0.96	1.00	0.93	1.00
4	10.00	1.00		0.96	1.00	0.93	1.00
5	9.00	0.89	*	0.84	0.95	0.82	0.96
6	10.00	1.00		0.96	1.00	0.93	1.00
7	10.00	1.00		0.96	1.00	0.93	1.00
8	7.45	0.72	*	0.62	0.79	0.59	0.84
9	8.73	0.86	*	0.77	0.91	0.74	0.81
10	10.00	1.00		0.96	1.00	0.93	1.00
11	8.18	0.80	*	0.70	0.86	0.67	0.88
12	10.00	1.00		0.96	1.00	0.93	1.00
13	7.91	0.77	*	0.67	0.83	0.64	0.85
14	7.91	0.77	*	0.67	0.83	0.64	0.85
15	7.55	0.73	*	0.63	0.80	0.60	0.82
16	7.55	0.73	*	0.63	0.80	0.60	0.82

* <0.90.

**Table 6 ijerph-19-02384-t006:** Qualitative evaluations by the experts.

Variables	No. of Contributions	Example	Action
2	4	It would be more convenient to indicate right side and left side of the court	The degree of openness has been changed to “player on the right side” and “player on the left side”
5	3	Please note the new WPT scoring system. “Golden point”	It has been indicated that if the tool is used to analyze matches in the WPT competition in the opening range of this variable, it would be modified, eliminating the option 40-advantage or advantage-40
8	5	This variable is very subjective. I think any point from a tie-break can be more key than a 40-0 from a first game of a set.	This variable was removed from the tool.
9	5	I don’t see it is interesting. It can give problems in the analysis. I see it as unnecessary.	This variable was removed from the tool.
11	6	Carry out a more specific degree of openness. “No lob” could be chiquita and passing.	The opening range has been changed to lob, chiquita and passing.
13	4	How is the observer going to differentiate an unforced error from a forced error? There are no unforced errors as there is rival opposition.	The degrees of opening in continuity, error and winning shot were redefined
14	4	How is the observer going to differentiate an unforced error from a forced error? There are no unforced errors as there is rival opposition.	The opening degrees were redefined as continuity, error, winning shot and no action
15	8	This variable does not depend only on the smash, but on many more actions. Justify opening ranges based on the scientific literature, by quartiles, by cluster ...	This variable was removed from the tool.
16	9	This variable does not depend only on the smash, but on many more actions. Justify opening ranges based on the scientific literature, by quartiles, by cluster ...	This variable was removed from the tool.

**Table 7 ijerph-19-02384-t007:** Reliability analysis of the NAPOA instrument.

		Adequacy	Wording	Total
Before	α	0.81	0.83	0.84
	Valid	16	16	32
After	α	0.90	0.88	0.89
	Valid	12	12	24

## Data Availability

Not applicable.

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
