# Peer review of "Analysis of the Actions of Net Zone Approach in Padel: Validation of the NAPOA Instrument"

_ijerph, 2022, doi:10.3390/ijerph19042384_

Round 1

Reviewer 1 Report

This article is well-organized with adequate results and resounding conclusions. The authors designed and validated a tool to ascertain the characteristics of the strokes that padel pairs use to reach the net and their consequences in the two subsequent shots of the game.

The topic discussed in the manuscript is attracting though I am not familiar with Padel. Authors aim to propose a validated tool instead of traditional observational methodology. However, this tool is still largely dependent on observations and expert's evaluations. Is it possible to fully construct a quantitative and more reliable model based on data via artificial intelligent approaches?

If so, please add some outlooks in the Discussion. The following papers that considered a deep-learning or other AI architecture applied in ice-hockey to detect players and moves, which might be helpful.

[1] Macdonald, B., 2012, March. An expected goals model for evaluating NHL teams and players. In Proceedings of the 2012 MIT Sloan Sports Analytics Conference, http://www. sloansportsconference. com.

[2] Schuckers, M. and Curro, J., 2013, April. Total Hockey Rating (THoR): A comprehensive statistical rating of National Hockey League forwards and defensemen based upon all on-ice events. In 7th annual MIT sloan sports analytics conference

Some minor points:

1) 'Advantaje' should be replaced with 'Advantage' in Table 2.

2) Should be 'No. of contributions' in Table 6. 

Author Response

The reviewer's proposal is very interesting. "Is it possible to fully construct a quantitative and more reliable model based on data via artificial intelligent approaches?"

However, we consider that in this case it is more appropriate to build a valid and reliable instrument but one that depends on the observations and evaluations of experts. There can be an infinity of different situations during this moment of the game that is intended to be studied. Game situations full of variability, technical shots, tactical and technical-tactical behaviors that must be observed, analyzed, interpreted and evaluated by experts.

The reviewer's corrections have been taken into account, as "advantaje" has been replaced by "advantage" in Table 2 and "Nº of contributions" by "No. of contributions in Table 6.

Reviewer 2 Report

Dear authors,

Your paper deals with an interesting topic and I believe that further improvements are need.

Section 1 is ok. You provided a well-grounded introduction and the study’s objective is clear.

Nevertheless, I believe that your paper needs a more strong theoretical background and a literature review section should be added.

At the end of the introduction section, you mentioned that you have reviewed of the scientific literature, yet the paper is missing it.

Furthermore, despite de characteristics of you study, I believe that to develop the measurement scale, you should follow in some degree the background literature. In subsection “2.4. Instrument” you made some advances in this issue, although I believe that a stronger theoretical emphasis could be made.

Section 4. is a bit descriptive and could be improved through a better emphasis in on breakthrough your module provides.

Section 5. Conclusions is a bit short and could be further extended. A better emphasis on the implications for theory and practice, the study limitations and future research could be added.

Good luck with your work

Author Response

We greatly appreciate your feedback.

Section 1. In the fourth paragraph of the introduction the scientific background of the analysis of the game in padel is explained and in the fifth the observation tools already designed on other aspects of the analysis of the game in padel are mentioned. All this information is compiled from the articles mentioned in the bibliography, which have been revised to write these paragraphs. Therefore, once these two paragraphs have been written, it is indicated below that the scientific literature has been reviewed and that, based on its review, the objective of the study is established.

The subsection "2.4. Instrument" details the scientific background and the explanation of all the variables of the instrument. Always relying on bibliographic references.

For all these reasons, we consider that the article has a very solid and adequate theoretical basis. In addition, it is always based on the scientific literature.

Section 4. We appreciate your comment, we consider that section 4 is appropriate.

Section 5. The comments on the conclusions are very pertinent, so they have been rewritten and expanded.

Reviewer 3 Report

The article is singular and it follows the scientific structure. The authors need to change or improve the next questions:

-Theoretical framework: too much long. Try to do more synthesis.

-Methods. It is well applied, but the election of the experts is not enough justified. The authors have to explain clearly and in detail the reason why of the election of the experts, and their profile. I suppose there is a request of anonymity by them, but, at least, they have to indicate the general data of them (company or university, merits). To obtain a whole article, I suggest to complete it with a Delphi.

-Results: it will be improved with the methodological new tools.

-Conclusions: too much brief. The authors have to expand it.

Author Response

We greatly appreciate your feedback.

- Theoretical framework: we consider it adequate. The introduction consists of six essential paragraphs. The first three deal with the current situation of paddle tennis at a social and scientific level. The fourth paragraph explains the scientific advances that have been developed on the subject of study, the analysis of the game in paddle tennis. Below are examples of other tools for the analysis of the game in padel and in the last paragraph the objective of the work is exposed.

- Methods and results: we consider that the information provided about the experts is sufficient and appropriate to know if they are qualified to participate as an expert in the study. 
Through the inclusion criteria established by the researchers:
i) to possess a Ph.D. ii) to possess the federative qualification as a trainer in padel and / or in another racket sport, iii) to teach or have taught at university, iv) to have publications with a theme oriented to the analysis of the game of padel and v) to work or have worked as a padel coach or coach of another racket sport.
We consider that it is not important and unnecessary to know other data about them.

- Conclusions: The comments on the conclusions are very pertinent and have therefore been rewritten and expanded.

Round 2

Reviewer 3 Report

The implementation of improvements in conclusions is right.